

# Arsenic and mercury tolerant rhizobacteria that can improve phytoremediation of heavy metal contaminated soils

Daniel Rojas-Solis, John Larsen and Roberto Lindig-Cisneros

Instituto de Investigaciones en Ecosistemas y Sustentabilidad, Universidad Nacional Autónoma de México, Morelia, Michoacan, Mexico

## ABSTRACT

**Background:** Mining deposits often contain high levels of toxic elements such as mercury (Hg) and arsenic (As) representing strong environmental hazards. The purpose of this study was the isolation for plant growth promoting bacteria (PGPBs) that can improve phytoremediation of such mine waste deposits.

**Methods:** We isolated native soil bacteria from the rhizosphere of plants of mine waste deposits and agricultural land that was previously mine tailings from Tlalpujahua Michoacán, Mexico, and were identified by their fatty acid profile according to the MIDI Sherlock system. Plant growth promoting traits of all bacterial isolates were examined including production of 3-indoleacetic acid (IAA), siderophores, biofilm formation, and phosphate solubilization. Finally, the response of selected bacteria to mercury and arsenic was examined an *in-vitro* assay.

**Results:** A total 99 bacterial strains were isolated and 48 identified, representing 34 species belonging to 23 genera. Sixty six percent of the isolates produced IAA of which *Pseudomonas fluorescens* TL97 produced the most. *Herbaspirillum huttiense* TL36 performed best in terms of phosphate solubilization and production of siderophores. In terms of biofilm formation, *Bacillus atrophaeus* TL76 was the best.

**Discussion:** Most of the bacteria isolates showed high level of tolerance to the arsenic (as $HAsNa_2O_4$ and $AsNaO_2$), whereas most isolates were susceptible to $HgCl_2$. Three of the selected bacteria with PGP traits *Herbispirillum huttiense* TL36, *Klebsiella oxytoca* TL49 and *Rhizobium radiobacter* TL52 were also tolerant to high concentrations of mercury chloride, this might could be used for restoring or phytoremediating the adverse environmental conditions present in mine waste deposits.

## INTRODUCTION

In the past few decades, soil pollution caused by heavy metals has increased in prevalence, especially in developing Countries (*Samuel et al., 2021*). Activities such as rapid industrialization, poor agricultural practices, and mine tailings runoff are often detrimental to soil health and can distribute hazardous metals into the environment, with harmful health consequences (*Raklami et al., 2022*).

Corresponding author
Roberto Lindig-Cisneros,
rlindig@cieco.unam.mx

In particular, the intensive extraction of minerals taking place in mining zones has produced a large volume of wastes and tailings, which release potentially toxic elements (PTE) to the environment, including heavy metals. Contamination with heavy metals, particularly arsenic (As) and mercury (Hg), is frequent throughout the planet (*Hering et al., 2017*).

Soil contamination by heavy metals has become a serious environmental issue, since most metals exert harmful effects at low concentrations (1–10 mg/mL), while some metals such as Hg have a toxic effect at even lower concentrations (0.001–0.1 mg/mL) also, being a mobile element in the environment, its negative impacts are considerable (*Kurniati et al., 2014*; *Devi et al., 2022*; *Zhang et al., 2010*). Arsenic pollution is not only a result of mining activities, but it also enters ecosystems using pesticide and/or herbicides based on this chemical element (*Kumar et al., 2016*).

One way of reducing the environmental impact of contaminated soils is the use of remediation strategies, some are chemical and physical in nature; however, these methods can be expensive and not always effective (*Kunito et al., 2001*). Biological techniques, such as phytoremediation and PGPB bioremediation, have emerged as environmentally sound approaches to heavy metal removal (*Devi et al., 2022*).

Microbial communities also play an important role during the recovery of soils by improving their structure and fertility and are excellent indicators because of their rapid response to environmental changes (*Xiao et al., 2017*). However, high concentrations of heavy metals might alter the diversity and structure of microbial communities (*Quadros et al., 2016*), their vital microbiological features (like growth, adhesion, and morphology), and essential biochemical traits such as respiration, nitrogen fixation, and mineralization of nitrogen and phosphorus (*Rajapaksha, Tobor-Kapłon & Bååth, 2004*). In consequence, soil microbial mediated phytoremediation requires choosing soil bacteria capable of growing in the presence of heavy metals, sometimes in high concentrations. Also, these bacteria have to show plant growth promotion features such as: synthesis of 3-indoleacetic acid (IAA), phosphate solubilization, biofilm formation, and production of siderophores that are important for plant growth in contaminated soils (*Khan et al., 2015*; *Yu et al., 2014b*; *Santoyo et al., 2016*).

In Tlalpujahua, Michoacán, located in west-central Mexico, an over 400-year history of mining activity in the region has generated tailings on which a partial natural regeneration of the vegetation has been taking place in the last 60 years since the mines in the region ceased to operate (*Corona-Chávez et al., 2010*). In the mining zone of Tlalpujahua-El Oro, little work has been carried out to remediate contaminated soils. *Osuna-Vallejo et al. (2019)*, evaluated the capability of the conifers *Juniperus deppeana* and *Pinus pseudostrobus* for extracting mercury from soils contaminated by mining deposits, and found that both species accumulated the metal in their wood, representing excellent candidates for phytoremediation by bioacumulation of mercury. PGPR can improve this type of phytoremediation processes (*Raklami et al., 2019*, *2021*).

The objective of this study was to select cultivable bacteria from tailing soils in Tlalpujahua, with traits of plant growth promotion and tolerance to mercury and arsenic for future use as inoculants in phytoremediation processes. We hypothesized that, in

altered soils and substrates (that result from long term mining), bacteria have evolved and developed tolerance mechanisms as a response to the selection pressure imposed by heavy metals, and some also have plant growth promotion mechanisms that make them potentially useful as phytoremediation agents. Our study is relevant due to the small number of studies that have been carried out in this area and the limitation of available strategies to restore soil conditions, so offering alternatives through native microorganisms represents an interesting biological restoration option.

## MATERIALS AND METHODS

### Site study and soil sampling

The municipality of Tlalpujahua is located in the state of Michoacán México, between the parallels 19°41′ and 19°53′ northern latitude, and 100°08′ and 100°18′ western longitude, at an elevation ranging from 2,200 to 3,100 m a.s.l. The characterization of our study site has previously been described by *Corona-Chávez et al. (2010, 2017)* and *Osuna-Vallejo et al. (2019)*, works that describe the physicochemical characteristics of soils in the Tlalpujahua-El Oro mining district and identified potentially toxic elements like arsenic (3.0–83.9 ppm), copper (63.8–548.2 ppm), lead (16.6–317.5 ppm) and mercury (20.51–110.7 ppm).

Three samples were collected from three different sites from the rhizosphere of native and cultivated plants at a depth of 0 to 20 cm: one in wetland dominated by *Typha domingensis* growing on a channel draining from an abandoned mine (N 19° 48′ 03.3″, W 100° 09′ 29.0″), a second from agricultural land planted with maize plants (N 19° 48′ 28.8″, W 100° 10′ 03.6″) and a third from another agricultural field also planted with maize (N 19° 48′ 26.8″, W 100° 10′ 00.2″; Fig. 1). The three samples were sieved through a 2 mm × 2 mm sieve to remove stones, debris, and other large particles, pooled, mixed, placed in sterile polyethylene bags, and stored in darkness at room temperature for the upcoming experiments.

Microorganisms were isolated from 0.5 g soil samples diluted in 1 mL of water, which were further diluted and plated on nutritive agar (BD Bioxon, Becto Dickinson de México) culturing medium in Petri dishes that were incubated 24 h at 30 °C. Ninety nine isolates were obtained based on colony morphology. The isolates were incubated at 28 °C for 24 h and preserved in 20% glycerol at −80 °C.

### Isolation of cultured rhizobacteria

The selected isolates were cultured in tryptic soy broth medium (TSB) and identified based on the Fatty Acid Methyl Ester (FAME) method based on their fatty acid profiles in the Microbial Identification System (MIDI) of the Sherlock System Software 4.0. The fatty acids were extracted by the four-step method (*Mansfeld-Giese, Larsen & Bödker, 2002*): (i) Saponification, (ii) methylation, (iii) extraction, and (iv) base wash. The fatty acid methyl esters were analyzed in an Agilent 6890 Plus gas chromatograph and identified in the Sherlock System Software 4.0 using the libraries recommended for aerobic heterotrophic bacteria (*Parsley, 1996*).

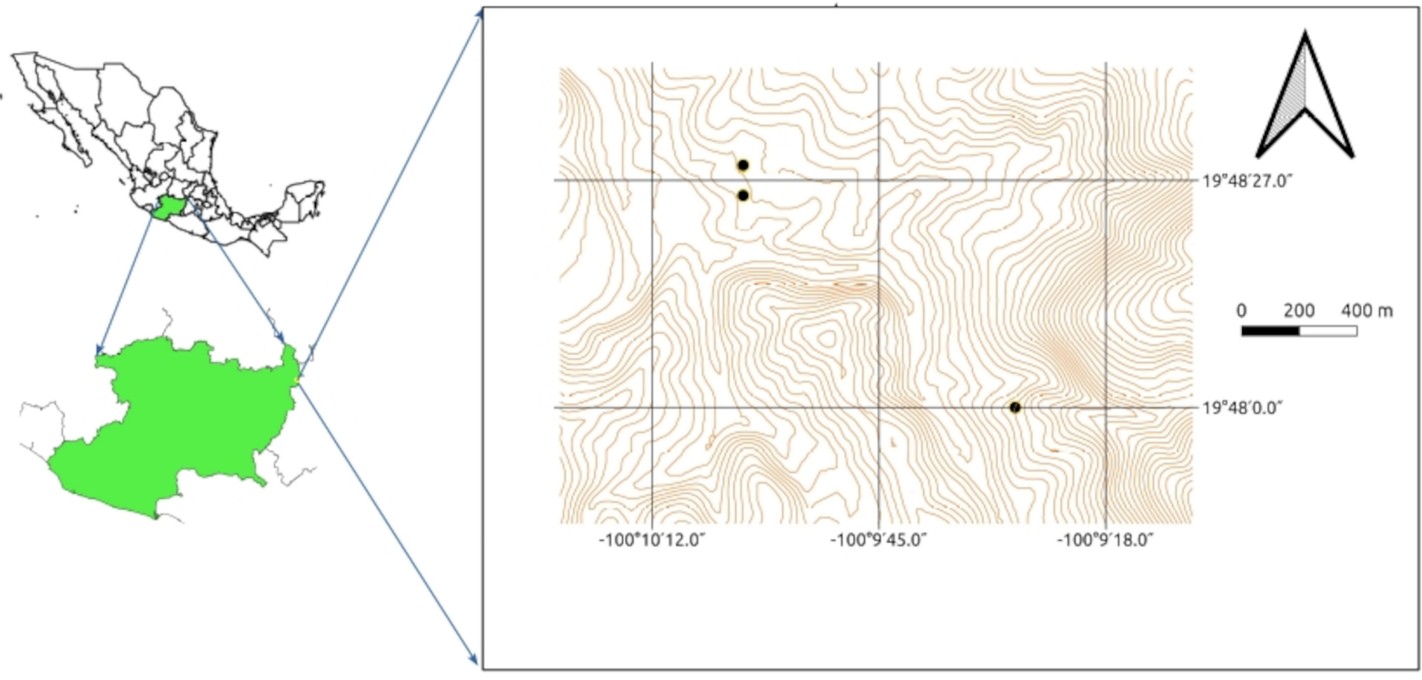

**Figure 1 Sampling sites within the Tlalpujahua-El Oro mining district in eastern Michoacán, México.** Each point indicates the site where soil was collected.

## Plant growth promoting traits

Production of siderophores was evaluated by triplicate in chrome azurol sulfonate (CAS) agar medium (*Santoyo, Sánchez-Yáñez & de los Santos-Villalobos, 2019*) cultures incubated at a temperature of 30 °C. The orange rings surrounding colonies were measured for 2 days.

The content of IAA was determined by the method described by *Patten & Glick (2002)*. Total of 25 mL flasks containing nutrient broth added with 1% tryptophan were inoculated and agitated in a rotating agitator at 150 rpm at a temperature of 30 °C. The cells were harvested by centrifugation at $10,000 \times g$ for 15 min and 2 mL of Salkowski reagent were added to the supernatant. The absorbance of the pink auxin complex was recorded at a wavelength of 540 nm in a Genesys 20 UV-VIS spectrophotometer. The standard curve was generated using dilutions of a pure IAA standard solution (Fluka, Buchs, Switzerland) and the uninoculated medium supplemented with Salkowski reagent as a control.

The biofilm forming capability of isolates was assessed in triplicate with at least two independent sets following the protocol of *Wei & Zhang (2006)*. Briefy, testing strains were grown in LB medium to an O.D. of 1 and then diluted (1:1,000) with fresh LB broth. A 0.5 ml diluted culture was transferred to an Eppendorf tube. Bacteria were incubated without agitation for 24, at 30 °C and the bioflm was quantifed.

Biofilm was stained with 0.1% crystal violet (w/v) for 15 min at room temperature and rinsed with water to remove loose cells and stain residues. The crystal violet stain was solubilized using 95% ethanol and the absorbance at 570 nm of the resultant stain solution was measured in a Genesys 20 UV-VIS spectrophotometer.

Finally, phosphate solubilization was determined in the Pikovskaya medium (with $Ca_3PO_4$ as phosphate source) in Petri dishes incubated at 30 °C and the yellow rings surrounding the colonies were measured daily for 2 days (*Chung et al., 2005*; *Joshi, Kumar & Brahmachari, 2021*).

### As-Hg minimal inhibitory concentration

Resistance of isolates to mercury and arsenic was assessed inoculating 10 μL of a $10^8$ cells/mL bacterial suspension in nutrient agar medium supplemented with sodium (meta) arsenite ($AsNaO_2$) and sodium arsenate dibasic ($HAsNa_2O_4$) solutions at concentrations of 200, 400, 600, 800, and 1,000 mg/kg, and mercuric chloride ($Cl_2Hg$) solution at concentrations of 23, 50, 75, 100, and 150 mg/kg. The minimal inhibitory concentration (MIC) was defined as the lowest concentration at which no colony units were formed after 3 days of incubation at a temperature of 30 °C. MIC for heavy metals was evaluated on solid media following studies that show the effectiveness of the procedure or the limitations of the traditional liquid MIC methods (*Yu et al., 2014a*; *Agarwal, Rathore & Chauhan, 2020*).

### Statistical analysis

Statistical analysis of the results was made in the software STATISTICA 8.0 applying one-way ANOVAs and the Duncan's multiple range test for mean comparison were used for multiple comparisons.

## RESULTS

Of the 99 isolates obtained, only 47 managed to grow in the presence of heavy metals. Seven isolates were obtained from site one, nineteen isolates from site two, and twenty-one isolates from site three. We identified a total of 34 species belonging to 23 genera. Table 1 shows cultivable bacteria obtained from three samples of rhizosphere soils from Tlalpujahua. From the first rhizospheric soil sample, the genera *Pseudomonas* and *Acinetobacter* were the most representative, appearing in two occasions, the predominant species were *Pseudomonas alcaligenes* and *Acinetobacter calcoaceticus*. From the second soil sample, the most common genus was *Bacillus*, as it was found on four occasions, two of these identified as *Bacillus*-GC group 22, while the species *Paracoccus denitrificans* was identified on two occasions. Finally, from the soil sample of site three, the most representative genera were *Bacillus* identified on three occasions and *Pseudomonas* on four occasions. The species that were identified in the greatest number of occasions were *Herbaspirullum huttiense* with two representatives and *Pseudomonas fluorescens* with three.

For the evaluation of the growth promotion characteristics of the 47 identified isolates, only one representative isolate was evaluated for each different species found (Table 2). Among the 27 strains tested nine strains did not produce IAA (therefore, they were not statistically compared with the producing strains), whereas the remaining 18 strains produced IAA in the range 2.13–131 mg $mL^{-1}$, production of IAA differed among strains ($F_{(17,36)}$ = 96.38; $P$ = 2 $e^{-16}$) Four strains produced more than 60 mg $mL^{-1}$ of the auxin:
**Table 1 Bacterial species isolated from three soil samples from the Tlalpujahua, Michoacán, Mexico.** Bacteria were identified based on their fatty acid profiles.

| Isolate | Source | Species | % Similarity (MIDI) |
|---------|--------|---------|---------------------|
| TL1 | Sample 1 | *Paucimonas lemoignei* | 0.303 |
| TL2 | Sample 1 | *Pseudomonas alcaligenes* | 0.677 |
| TL3 | Sample 1 | *Citrobacter amalonaticus* | 0.471 |
| TL4 | Sample 1 | *Acinetobacter calcoaceticus* | 0.708 |
| TL6 | Sample 1 | *Pseudomonas alcaligenes* | 0.473 |
| TL7 | Sample 1 | *Acinetobacter calcoaceticus* | 0.512 |
| TL8 | Sample 1 | *Herbaspirillum huttiense* | 0.355 |
| TL11 | Sample 2 | *Stenotrophomonas maltophilia* | 0.278 |
| TL21 | Sample 2 | *Sphingobacterium faecium* | 0.378 |
| TL22 | Sample 2 | *Pedobacter heparinus* | 0.221 |
| TL23 | Sample 2 | *Paracoccus denitrificans* | 0.840 |
| TL24 | Sample 2 | *Rhodobacter sphaeroides* | 0.747 |
| TL30 | Sample 2 | *Paracoccus denitrificans* | 0.901 |
| TL31 | Sample 2 | *Cellulomonas fimi*-GC subgroup A | 0.613 |
| TL33 | Sample 3 | *Brevibacillus centrosporus* | 0.558 |
| TL34 | Sample 3 | *Bacillus alcalophilus* | 0.568 |
| TL35 | Sample 3 | *Brevibacillus choshinensis* | 0.681 |
| TL36 | Sample 3 | *Herbaspirillum huttiense* | 0.496 |
| TL37 | Sample 3 | *Kocuria rosea*-GC subgroup A | 0.768 |
| TL38 | Sample 3 | *Pseudomonas fluorescens* | 0.582 |
| TL39 | Sample 3 | *Bacillus megaterium* | 0.639 |
| TL40 | Sample 3 | *Paenibacillus validus* | 0.432 |
| TL41 | Sample 3 | *Staphylococcus gallinarum*-GC subgroup A | 0.597 |
| TL43 | Sample 3 | *Paenibacillus alvei*-GC subgroup A | 0.454 |
| TL44 | Sample 3 | *Microbacterium barkeri* | 0.819 |
| TL47 | Sample 3 | *Paenibacillus pabuli* | 0.656 |
| TL48 | Sample 3 | *Brevundimonas vesicularis* | 0.771 |
| TL49 | Sample 3 | *Klebsiella oxytoca*-GC subgroup B | 0.528 |
| TL51 | Sample 3 | *Bacillus*-GC group 22 | 0.551 |
| TL52 | Sample 2 | *Rhizobium radiobacter* | 0.811 |
| TL54 | Sample 2 | *Rhodococcus wratislaviensis* | 0.706 |
| TL55 | Sample 2 | *Microbacterium barkeri* | 0.695 |
| TL62 | Sample 2 | *Arthrobacter globiformis*-GC subgroup A | 0.708 |
| TL64 | Sample 2 | *Nocardia otitidiscaviarum* | 0.477 |
| TL65 | Sample 2 | *Paenibacillus polymyxa* | 0.563 |
| TL67 | Sample 2 | *Pantoea agglomerans* | 0.344 |
| TL68 | Sample 2 | *Bacillus*-GC group 22 | 0.707 |
| TL69 | Sample 2 | *Bacillus*-GC group 22 | 0.587 |
| TL74 | Sample 2 | *Sphingopyxis macrogoltabida* | 0.894 |
| TL76 | Sample 2 | *Bacillus atrophaeus* | 0.492 |
| TL79 | Sample 2 | *Bacillus alcalophilus* | 0.568 |

| | Table 1 (continued) | | |
|---|---|---|---|
| **Isolate** | **Source** | **Species** | **% Similarity (MIDI)** |
| TL80 | Sample 3 | *Pseudomonas putida* | 0.161 |
| TL84 | Sample 3 | *Stenotrophomonas maltophilia* | 0.649 |
| TL85 | Sample 3 | *Rhizobium rubi* | 0.645 |
| TL86 | Sample 3 | *Klebsiella oxytoca*-GC subgroup B | 0.401 |
| TL91 | Sample 3 | *Pseudomonas fluorescens* | 0.634 |
| TL93 | Sample 3 | *Herbaspirillum huttiense* | 0.492 |
| TL97 | Sample 3 | *Pseudomonas fluorescens* | 0.907 |

*Sphingopyxis macrogoltabida* TL74 ($64.28$ mg mL$^{-1}$), *H. huttiense* TL36 ($73.40$ mg mL$^{-1}$), and *Pseudomonas putida* TL80 ($74.20$ mg mL$^{-1}$). The strain showing the highest auxin production was *P. fluorescens* TL97 ($131.02$ mg mL$^{-1}$).

Concerning phosphate solubilization 15 strains were negative, whereas the remaining 12 strains showed statistically different levels of activity ($F_{(11,36)} = 22.15$; $P = 8.42$ e$^{-13}$). Highest phosphate solubilization activity was observed for *H. huttiense* TL36 (halo diameter of $18.71$ mm), *Staphylococcus gallinarum* TL41 ($15.25$ mm), and *Rhizobium radiobacter* TL52 ($14.87$ mm).

Except for *Bacillus alcalophilus* TL79, all examined strains produced siderophores, being the differences statistically significant ($F_{(26,54)} = 24.24$; $P = 2$ e$^{-16}$). The largest halo was observed with *Bacillus atrophaeus* TL76 with $10.33$ mm in diameter, *A. calcoaceticus* TL7 with $11.66$, and *H. huttiense* TL36 with $17.6$ mm.

Biofilm formation capacity for the 27 isolates varied considerably between strains within the range ($0.05$ and $0.21$), being the differences statistically significant ($F_{(26,54)} = 9.74$; $P = 1.72$ e$^{-12}$). Four strains had the highest activity: *B. atrophaeus* TL76 (optical absorption at 570 nm of $0.21$), *Paucimonas lemoignei* TL1 ($0.20$), *R. radiobacter* TL52 ($0.18$), and *A. calcoaceticus* TL7 ($0.17$).

Based on the performance shown in the evaluation of growth promotion and bioremediation traits, eleven isolates were selected to determine their tolerance to different concentrations of arsenic and mercury (Table 3). All selected strains tolerated the maximum tested concentration ($1,000$ mg kg$^{-1}$) of HAsNa$_2$O$_4$ and were able to grow in that condition, while only nine of the isolates could grow in that same concentration of AsNaO$_2$. Highest MICs for AsNaO$_2$ were observed for *P. fluorescens* TL97 and *P. putida* TL80, with $400$ mg kg$^{-1}$ and $600$ mg kg$^{-1}$, respectively. Six strains were unable to grow at the lowest concentration of Cl$_2$Hg tested ($23,400$ mg kg$^{-1}$), two strains (TL23 and TL97) reached their MIC value at $75,400$ mg kg$^{-1}$, and four strains (TL36, TL49, TL52, and TL80) had a MIC of $100$ mg kg$^{-1}$.

## DISCUSSION

One the alternatives for mine tailings or contaminated soil restoration is the use of phytoremediation in combination with inoculation with PGPBs, which, through alteration of the mobility and bioavailability of the metals, play an essential role in facilitating plant

**Table 2 Plant growth promoting features.** Summary of plant growth promoting features displayed by the selected isolates.

| Bacteria | IAA[†] | Phosphate solubilization[††] | Siderophores[††] | Biofilm production[†††] |
|---|---|---|---|---|
| *Paucimonas lemoignei* TL1 | 28.99 ± 2.38cd | 6.25 ± 0.25ef | 5.33 ± 0.33efg | 0.20 ± 0.01ab |
| *Pseudomonas alcaligenes* TL2 | 27.99 ± 0.20cd | 10.25 ± 1.18cd | 5.6 ± 0.60ef | 0.10 ± 0.02cde |
| *Acinetobacter calcoaceticus* TL7 | 40.15 ± 0.24de | ND | 11.66 ± 1.30b | 0.17 ± 0.40ab |
| *Sphingobacterium faecium* TL21 | ND | 7.12 ± 0.98ef | 4.83 ± 0.33efg | 0.05 ± 0.007ef |
| *Paracoccus denitrificans* TL23 | 8.39 ± 0.17a | 7.12 ± 0.47ef | 4.5 ± 0.28efgh | 0.075 ± 0.003def |
| *Cellulomonas fimi*-GC subgroup A TL31 | ND | 6 ± 0.35e | 3.33 ± 0.16gh | 0.069 ± 0.005def |
| *Brevibacillus centrosporus* TL33 | ND | ND | 5 ± 0.28efg | 0.060 ± 0.006ef |
| *Brevibacillus choshinensis* TL35 | 2.13 ± 0.07[a] | 5.25 ± 0.14e | 5.66 ± 0.88ef | 0.069 ± 0.01def |
| *Herbaspirillum huttiense* TL36 | 73.40 ± 2.56g | 18.87 ± 1.65a | 17.6 ± 0.60a | 0.088 ± 0.004def |
| *Kocuria rosea*-GC subgroup A TL37 | 22.08 ± 1.59bc | ND | 4 ± 0.28fgh | 0.087 ± 0.01def |
| *Paenibacillus validus* TL40 | 2.92 ± 0.22a | ND | 8.66 ± 0.16cd | 0.099 ± 0.01def |
| *Staphylococcus gallinarum*-GC subgroup A TL41 | ND | 15.25 ± 1.88b | 5 ± 0.57efg | 0.092 ± 0.01def |
| *Paenibacillus alvei*-GC subgroup A TL43 | ND | ND | 8.33 ± 1.16d | 0.082 ± 0.006def |
| *Microbacterium barkeri* TL44 | ND | ND | 3.83 ± 0.44fgh | 0.070 ± 0.008def |
| *Paenibacillus pabuli* TL47 | 22.01 ± 0.40bc | ND | 2.66 ± 0.16h | 0.12 ± 0.01cd |
| *Brevundimonas vesicularis* TL48 | ND | ND | 4.16 ± 0.16fgh | 0.06 ± 0.004ef |
| *Klebsiella oxytoca*-GC subgroup B TL49 | 19.27 ± 0.40b | 9.12 ± 0.42cde | 6 ± 1ef | 0.087 ± 0.01def |
| *Rhizobium radiobacter* TL52 | 34.20 ± 1.02de | 14.87 ± 0.94b | 5.5 ± 1efg | 0.183 ± 0.02ab |
| *Rhodococcus wratislaviensis* TL54 | ND | ND | 4.5 ± 0.57efgh | 0.117 ± 0.01cd |
| *Arthrobacter globiformis*-GC subgroup A TL62 | 46.49 ± 0.62e | ND | 10 ± 0.76c | 0.070 ± 0.001def |
| *Paenibacillus polymyxa* TL65 | 8.54 ± 1.17a | 7.5 ± 0.20def | 4 ± 0.28fgh | 0.155 ± 0.004bc |
| *Bacillus*-GC group 22 TL68 | 3.93 ± 0.06a | ND | 4.83 ± 0.88fgh | 0.083 ± 0.003def |
| *Sphingopyxis macrogoltabida* TL74 | 64.28 ± 6.78f | ND | 4.86 ± 0.46fgh | 0.105 ± 0.008def |
| *Bacillus atrophaeus* TL76 | ND | ND | 10.33 ± 0.97c | 0.21 ± 0.01a |
| *Bacillus alcalophilus* TL79 | 18.14 ± 1.85b | ND | ND | 0.005 ± 0.007ef |
| *Pseudomonas putida* TL80 | 74.20 ± 5.99h | 11.12 ± 0.31c | 4.33 ± 0.60fgh | 0.086 ± 0.008def |
| *Pseudomonas fluorescens* TL97 | 131. 02 ± 10.08i | ND | 6.5 ± 0.57bc | 0.070± 0.009def |

Notes:
[†] IAA production measured in mg/L.
[††] Measured as diameter in mm of the ring of solubilization surrounding the colonies.
[†††] Measured as absorbance at 570 nm.
Lowercase letters indicate significant differences relative to the control in Duncan's multiple range tests ($P < 0.05$). ND, Not determined.

growth in conditions of high heavy metal concentrations in the soil (*Rehman et al., 2019*; *Tara et al., 2019*; *Yahaghi et al., 2019*). Currently, many microbial genera have been reported for their ability to reduce the toxic effects caused by heavy metals in the environment, being the most frequent; *Aeromonas, Rahnella, Ochrobactrum, Microbacterium, Azospirillum, Rhizophagus, Klebsiella, Enterobacter, Ralstonia, Rhizobium, Bacillus*, and *Pseudomonas* (*Mishra, Singh & Arora, 2017*; *Tiwari & Lata, 2018*).

**Table 3 Torelance to heavy metals.** Tolerance to the heavy metals, arsenic and mercury, of isolates growing in the presence of heavy metal salts ((+) Growth of colonies, (−) No growth of colonies).

|  | TL 1 | TL 2 | TL 7 | TL 23 | TL 35 | TL 36 | TL 49 | TL 52 | TL 65 | TL 80 | TL 97 |
|---|---|---|---|---|---|---|---|---|---|---|---|
| $HAsNa_2O_4$ |  |  |  |  |  |  |  |  |  |  |  |
| 200 mg/kg | + | + | + | + | + | + | + | + | + | + | + |
| 400 mg/kg | + | + | + | + | + | + | + | + | + | + | + |
| 600 mg/kg | + | + | + | + | + | + | + | + | + | + | + |
| 800 mg/kg | + | + | + | + | + | + | + | + | + | + | + |
| 1,000 mg/kg | + | + | + | + | + | + | + | + | + | + | + |
| $AsNaO_2$ |  |  |  |  |  |  |  |  |  |  |  |
| 200 mg/kg | + | + | + | + | + | + | + | + | + | + | + |
| 400 mg/kg | + | + | + | + | + | + | + | + | + | + | − |
| 600 mg/kg | + | + | + | + | + | + | + | + | + | − | − |
| 800 mg/kg | + | + | + | + | + | + | + | + | + | − | − |
| 1,000 mg/kg | + | + | + | + | + | + | + | + | + | − | − |
| $Cl_2Hg$ |  |  |  |  |  |  |  |  |  |  |  |
| 23 mg/kg | − | − | − | + | − | + | + | + | − | + | + |
| 50 mg/kg | − | − | − | + | − | + | + | + | − | + | + |
| 75 mg/kg | − | − | − | − | − | + | + | + | − | + | − |
| 100 mg/kg | − | − | − | − | − | − | − | − | − | − | − |
| 150 mg/kg | − | − | − | − | − | − | − | − | − | − | − |

Molecular studies such as metagenomics that allow us to study genetic material from environmental samples are extremely useful (*Datta et al., 2020*). However, the use of cultivable microorganisms allows us to identify and reproduce bacteria with potentially useful characteristics for phytoremediation by promoting plant growth.

In this study, we applied a culture-dependent approach which consisted in the identification of microbial communities based on the groupings of fatty acids. The FAME method allows to differentiate the main taxonomic groups within a community (*Kirk et al., 2004*). However, one of the limitations of this technique is that in some cases the differences between fatty acid profiles cannot be contrasting enough to accurately establish the different species that make up a community (*Bing-Ru et al., 2006*). Being aware of this limitation, we identified 34 species of bacteria belonging to 23 genera. It is important to consider that the diversity of bacterial groups in soils is conditioned by soil conditions and by the identification techniques, so our data might under-represent the true diversity or the sampled soils.

The most abundant identified genera were *Bacillus, Pseudomonas, Paenibacillus, Herbaspirillum*, and *Acinetobacter*. Using culture-dependent methods, *Hamood et al. (2020)* found eleven isolates tolerant to arsenic that were native to gold mining tailings in Malaysia, among which the dominant genera were *Bacillus, Pseudomonas, Lysinibacillus,* and *Micrococcus*. The first two also found in our study.

In a study of the bacterial community growing in soils contaminated with heavy metals, *Tipayno et al. (2018)* found that the structure of the community at the level of phylum depended on the general soil properties. At lower taxonomic levels, the concentrations of arsenic and lead were significant; and species of the genus *Bacillus* were positively correlated with the concentration of arsenic. Using a culture-independent metagenomic approach, *Hemmat-Jou et al. (2018)* addressed the biodiversity of the microbial community in soils contaminated with lead and zinc and found ten most abundant bacterial genera: *Solirubrobacter, Geobacter, Edaphobacter, Pseudomonas, Gemmatiomonas, Nitrosomonas, Xanthobacter, Sphingomonas, Pedobacter,* and *Ktedonobacter*. This study differs from ours in the contaminating heavy metals and the identification methodology. However, in general, the latter previous reports agree with our results in that *Bacillus* and *Pseudomonas* predominate in soils contaminated with heavy metals and with the review by *Fakhar et al. (2020)*, that concluded that *Bacillus* and *Pseudomonas* are two of the most frequent genera responsible of bioremediation of contaminated soils.

The presence of microorganisms with bioremediation and phytoremediation potential by producing plant growth promotion compounds, are essential for the search of strategies for the for restoration of soils contaminated with heavy metals. In our study, the isolate with the best plant growth promotion and heavy metal tolerance was *H. huttiense* TL36, outstanding for its performance in phosphate solubilization and siderophore production.

Siderophore producing bacteria play a major role in the survival and growth of plants present in tailings soils by alleviating the toxicity of metals and providing nutrients, which is due to the combination of bacterial siderophores with metals other than iron, that might explain why microorganisms are able to survive in tailings contaminated with heavy metals (*Adler et al., 2012*). As mentioned above, the TL36 isolate stood out for its ability to solubilize phosphates. In habitats such as mine tailings, the ability of microorganisms to solubilize recalcitrant substances such as phosphorus is a trait that determines their ability to adapt to these environments (*Jones & Oburger, 2011*).

Additionally, phosphate solubilizing bacteria in these environments help in the establishment of vegetation. It is known that bacteria can reduce $HAsNa_2O_4$ to $AsNaO_2$ the latter accumulates in aerial parts of plants. To overcome the stress caused by arsenic, plants require an adequate supply of phoshorus, therefore bacteria capable of phosphate solubilization are essential for an adequate remediation of soil contamination with $HAsNa_2O_4$ (*Alka et al., 2020*).

PGPBs not only mitigate metal toxicity but also act as plant growth promoters. The production of IAA emitted by bacteria alters various physiological processes in plants related with stress tolerance and growth. *Ahemad & Kibret (2014)* reported that IAA, in addition to stimulating root growth, facilitates water movement, controls vegetative growth, and initiates formation of adventitious and lateral roots. In our study the best producers of IAA were *P. fluorescens* TL97 (131.02 mg/L) and *P. putida* TL80 (74.20 mg/L). IAA producing bacteria have proven their usefulness due to their role in plant-bacteria interactions and plant growth in heavy metal contaminated soils (*Chiboub et al., 2016*). Previous reports have highlighted the species in the genus *Pseudomonas*

particularly *P. fluorescens* and *P. putida* as outstanding producers of IAA (*Saharan & Nehra, 2011*).

In a review of the role of exopolysaccharides in metal removal, *Gupta & Diwan (2017)* describe the mechanisms by which biofilms can immobilize or modify the redox state of metals, thus reducing their toxic effects on plants. Therefore, this ability is important for the restoration of soils contaminated with heavy metals. Our results showed that the best biofilm producer was *Bacillus atrophaeus* TL76. Several species in the genus *Bacillus* have the capability of responding to stress by producing biofilm (*Bais, Fall & Vivanco, 2004*; *Kasim et al., 2016*). Biofilm production becomes essential for bacteria growing in heavy metal contaminated soils. Because biofilms are negatively charged, they can adhere to surfaces by electrostatic attraction and to normally positively charged heavy metals, the biofilm layered bacteria thus acting as biosorbents of metals in the soil, therefore reducing their bioavailability to plants (*Kalita & Joshi, 2017*).

Because tolerance to heavy metals is one of the key factors of bioremediation and restoration of tailings soils using microorganisms, we evaluated tolerance to As (III), As (V), and Hg (I) in eleven isolates selected for their performance in assays of plant growth promotion and bioremediation features. If the bacteria isolated from tailings soils in Tlalpujahua are adapted to withstand heavy metal contamination in soils, we demonstrated their heavy metal tolerance. All assayed isolates showed adaptation to the presence of 1,000 mg kg$^{-1}$ of pentavalent arsenic ($HAsNa_2O_4$), possibly due to associated mechanisms including bioaccumulation, oxidation-reduction reactions, efflux mechanisms, and others (*Hamood et al., 2020*).

In the presence of trivalent arsenic ($AsNaO_2$), the only isolates that did not tolerate the medium concentration we tested (600 mg/kg) were *P. putida* TL80 and *P. fluorescens* TL97. The toxicity of As (III) is tenfold that of As (V), which is due to the former reacting with thiols of small molecules and sulfhydryl residues of cysteine in proteins thus inhibiting essential biochemical processes in organisms including bacteria (*Rosen & Liu, 2009*). *Román-Ponce et al. (2018)* isolated 27 bacterial strains from the rhizosphere of *Prosopis laevigata* and *Sphaeralcea angustifolia* growing in mine tailings in Santa María, San Luis Potosí, Mexico. The authors determined the minimum inhibitory (MIC) concentration of the isolates that they identified as belonging to the genera *Arthrobacter*, *Bacillus*, *Brevibacterium*, *Kocuria*, *Microbacterium*, *Micrococcus*, *Pseudomonas*, and *Staphylococcus* by culturing them in variable concentrations of arsenic, finding MIC values from 20 to over 100 mM for As (V) and between 10–20 mM for As (III), corroborating that the latter has toxic effects limiting bacterial growth.

In our assays for tolerance to $HgCl_2$, none of the tested isolates was able to tolerate the minimum concentration we used (200 mg/kg). The toxic effect of relatively low concentrations of mercury that we observed on bacteria isolated from soils in Tlalpujahua agrees with the results of *Harries-Hellal et al. (2009)* the first to report the consequences on the soil microbiota of the presence of mercury, who observed that 0.1 mg kg$^{-1}$ of mercury changed the structure and abundance of the microbial soil communities, a change that became more noticeable at a concentration of 20 mg kg$^{-1}$.

The performance of the isolates *H. huttiense* TL36, *K. oxytoca* TL49, and *R. radiobacter* TL52 in our heavy metal tolerance assays was outstanding. The tolerance of species in the genera *Rhizobium* and *Kleibsella* to several heavy metals had previously been recognized by authors like *Deepika et al. (2016)*, *Meena et al. (2018)*, *Mohan et al. (2019)*, *Kumar et al. (2021)*, and *Chakraborty et al. (2021)*.

Some species of the genus *Klebsiella* have been isolated from contaminated soils, and it has been reported that strains of *K. pneumoniae* y *K. Oxytoca* tolerate high concentration of cadmium and arsenic (*Shakoori et al., 2010*; *Shamim & Rehman, 2012*). *Kumar et al., 2021*, obtained 108 isolates of arsenic resistant bacteria from mining sites in India. Among them strain RnASA11 of *Klebsiella pneumonía* was resistant to 600 mM As (V) y 30 mM As (III), and was capable under controlled conditions of reducing the concentration of arsenate in 44% and arsenite in 38.8% when compared with a control.

*Rhizobium* has been studied mostly for its nitrogen fixation and symbiotic capabilities (*Masson-Boivin & Sachs, 2018*), but its role as a bioindicator of heavy metal presence in the soil has been also studied (*Stan et al., 2011*). *Deepika et al., 2016*, isolated *Rhizobium* from nodules of *Vigna radiata*. Isolate VBK102 that was identified as *Rhizobium radiobacter*, produced exopolysacharides that sequestered arsenic (10% of total cell weight). This species tolerates several heavy metals such as: As(V) (10 mM), Cu (1.5 mM), Pb (0.18 mM), Cr (0.1 mM), Ni (0.08 mM), and Cd (0.004 mM).

Finally, althogh *Herbaspirrillum* is not frequent in heavy metal contaminated soils, *Govarthanan et al. (2014)*, found that *Herbaspirillum* sp. GW103 played a role in the biolixiviation of copper in mine deposits. This strain (GW103) besides its effect on Cu, was resistant to As (550 mg/l), Cu (350 mg/L), zinc (Zn) (300 mg/L) and plomo (Pb) (200 mg/L).

## CONCLUSIONS

Throughout the world, various anthropogenic activities, in particular mining, have directly or indirectly deposited enormous amounts of PTEs in the soil, resulting in negative consequences for the environment and human health. Consequently, identifying bacterial strains that can improve phytoremediation or could facilitate the establishment of native vegetation in the generally uncovered tailing soils offers a promising option for their restoration. In this study, we found genera that have been widely reported in soils contaminated with heavy metals, such as *Bacillus* and *Pseudomonas*, also bacteria that are resistant to Hg and As and at the same time present plant growth promotion properties such as *H. huttiense* TL36, *R. radiobacter* TL52 and *K. oxytoca* TL49. It is important to mention that *Herbaspirillum* is reported for the first time as a genus that has tolerance to high concentrations of heavy metals and participates in the solubilization of phosphates. The identification using the FAME method allowed us to have an approach to the isolated species in our study, however, in future perspectives these isolates that were promising will be identified through phylogenetic analyzes based on the 16SrRNA gene. Our results suggest that isolates TL36, TL52 and TL49 can be an excellent alternative in the remediation of mine tailings; however, some tests are still needed to evaluate the ability of

these isolates to remove or transform heavy metals that allow us to corroborate their potential as bioremediation agents.

Additionally, the advantage of obtaining cultivable microorganisms allows us to think of the use of inoculants as a plant-microorganism interaction strategy to optimize bioremediation processes. Therefore, in future perspectives, isolates TL36, TL52 and TL49 will be inoculated in representative crops of the municipality of Tlalpujahua to validate their results under greenhouse conditions as growth promotion and bioremediation agents.

### Funding
Daniel Rojas-Solis received a postdoctoral scholarship from DGAPA-UNAM. This research was funded by research grant IG200221 by DGAPA-UNAM. The funders had no role in study design, data collection and analysis, decision to publish, or preparation of the manuscript.

### Grant Disclosures
The following grant information was disclosed by the authors:
DGAPA-UNAM: IG200221.

### Competing Interests
The authors declare that they have no competing interests.

### Author Contributions
- Daniel Rojas-Solis conceived and designed the experiments, performed the experiments, analyzed the data, prepared figures and/or tables, authored or reviewed drafts of the article, and approved the final draft.
- John Larsen analyzed the data, authored or reviewed drafts of the article, and approved the final draft.
- Roberto Lindig-Cisneros conceived and designed the experiments, authored or reviewed drafts of the article, and approved the final draft.

### Data Availability
The raw data shows all results on bacteria growth in response to treatments and fatty acid profile results and is available in the Supplemental File.

### Supplemental Information
Supplemental information for this article can be found online at http://dx.doi.org/10.7717/peerj.14697#supplemental-information.

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
