# Peer review of "Arsenic and mercury tolerant rhizobacteria that can improve phytoremediation of heavy metal contaminated soils"

_PeerJ, doi:10.7717/peerj.14697_

## Round 0.1 · original submission · Major Revisions

Dear Dr. Rojas-Solis and colleagues:

Thanks for submitting your manuscript to PeerJ. I have now received three independent reviews of your work, and as you will see, the reviewers raised some concerns about the research. Despite this, these reviewers are optimistic about your work and the potential impact it will lend to research on plant growth promoting traits and As-Hg tolerance of rhizobacteria from mines. Thus, I encourage you to revise your manuscript, accordingly, taking into account all of the concerns raised by the reviewers.

Please improve the content and clarity of the figures and tables (see suggestions by the reviewers). Please also ensure that all appropriate references are included. Overall, the presentation of the manuscript needs improvement, but the authors have provided plenty of suggestions that I feel will greatly improve your manuscript.

Reviewers 1 & 3 have requested that you cite specific references. You may add them if you believe they are especially relevant. However, I do not expect you to include these citations, and if you do not include them, this will not influence my decision.

I look forward to seeing your revision, and thanks again for submitting your work to PeerJ.

Good luck with your revision,

-joe

·

Basic reporting

Some of the useful research papers to enhance the journal standard Try to include it in your revised manuscript.

1. https://link.springer.com/article/10.1007/s13205-013-0140-6
2. https://link.springer.com/article/10.1007/s10311-020-01081-y
3. https://link.springer.com/article/10.1007/s10311-020-01010-z
4.https://www.sciencedirect.com/science/article/abs/pii/S0048969721008962
5. https://www.sciencedirect.com/science/article/abs/pii/S0045653521018403
6.https://www.sciencedirect.com/science/article/abs/pii/S0304389421025401

In lines 130-134, materials and methods, unrelevant information are kept. remove those lines.
In line 144, what do you mean by nutritive agar?. Normally bacteria will grow in 24 hours. But why 3 days incubation was kept?
line 146, cultured several times until 99 pure cultures were obtained. Make it more clear
Rewrite the sentence from line number 167-170
There is no proof of evidence for the IAA , Auxin, and bioflim formation. Authors are request to keep the images for additional support.

Experimental design

Instead of giving everyting in the tabular column format, authors can give it in bar chart

Validity of the findings

What is the novelty of this work?

Additional comments

Rewrite the manuscript with english expert . Many typo errors are found in the text.

Reviewer 2 ·

Basic reporting

The authors need to revise some of the contents in terms of clear and unambiguous english. No sufficient and clear background of the study is provided. The presentation of data, results and figures needs refinement to enhance the readability and clarity. There is no clear aim of the study proposed. Authors must include a section on "rationale of the study and key highlights" at the end of the Introduction section.

Experimental design

1. The rationale of isolating bacteria from soil using nutrient media is not clear. It would have been relevant if bacteria could have been enriched on minimal media putting mine soil as enrichment, then isolate on MSM media with specific condition given.
2. I am unable to understand the logic of isolating only heterotrophic bacterial members from a metal-impacted soil site which might harbour lithotrophic to chemotrophic members and it is most relevant to study their metabolic intricacies/inventories.
3. “cultured several times until 99 pure cultures were obtained”…I could not understand the meaning of this phrase.
4. Siderophore and phosphate solubilization were estimated qualitatively…but others were quantified…Why? Authors should maintain uniformity in assessing all the traits. It would be relevant if authors report on: a) growing strains on MSM with relevant carbon sources (similar to site) and check for IAA production, b) phosphate solubilization and others..
6. MIC for heavy metals was evaluated on solid media. This is not the standard way. It shuld be done using tube dilution method. Authors should do the test and present the data.
7. Authors have not characterized the site, for example total metal content, soil characteristics, organic content, etc. That would give better picture of the sampling site.
8. Why the authors have not performed metal transformation or detoxification tests? Because simple tests for metal tolerance by bacteria does not indicate anything relevant for using these microbes for mine soil remediation.
9. More statistical test could have been done to cluster the strains based on their performance and correlate with their applicability.

Validity of the findings

The background of the study has not been stated adequately. Though most mining activities leads to contamination of various heavy metals, but these contaminants are mine specific. For example, coal, copper and thorium.. So, authors should be more specific in this regard. Why authors did not search for metal detoxifying or bio accumulating (removal) microbes?
Why authors identified through MIDI Sherlock? It is not the standard best practices of robust identification? MIDI is being recommended for FAME analyses for species demarcation. It would have been better if full-length 16S rRNA gene or conserved house-keeping gene based phylogenetic analyses could have performed.
Why response of microbes to As and Hg was only checked? Why not other relevant heavy metals?
The results and discussion presented seems flat and uninteresting.
“According to FAO (2015)……..”..The sentence reads shabby and needs rephrasing. Besides, I could not find FAO (2015) in reference list. Please be careful.
Instead of long literature survey on various heavy metals from diverse mines and microbes tolerating these metals, authors should present straightway on their study site, Tlalpujahua, Michoacn, located in west-central Mexico. The background looks lengthy and irrelevant. Please rephrase.
In the last paragraph of introduction, authors stated that the aim was to assess the culture-dependent methods for isolating bacterial strains…..That’s not correct. Authors should clearly state the aim of the study.
Overall the results are not well presented. I find result as data values.
Regarding discussion, authors have discussed in a lengthy way, which are not justified. I was got lost while reading the results and discussion. Please rephrase and re-write to bring relevant points reated to the results obtained.

Additional comments

Authors should present future prospects of the study and knowledge gap.
The results can be well presented through data figures instead of through tables only.

Reviewer 3 ·

Basic reporting

The revised manuscript entitled “Characterization of native plant growth promoting bacteria of mine waste deposit soils” investigate the plant growth promoting traits and As-Hg tolerance of rhizobacteria isolated form one mine site and two agriculture sites. To reach their objectives, the author did an interesting works. The objectives proposed by the authors are potentially interesting and of considerable importance. However, there are too many raised issues that should be addressed to improve the paper quality.

Experimental design

Title
- The title does not reflect the core of the study since only 15% of the isolated belong to the contaminated sites, please modify the title to properly reflect the results of the whole study. I suggest: As and Hg tolerant rhizobacteria as a strategy to remediate heavy metals contaminated soils.
Abstract
- L19: Please replace “search” by “isolation”.
- L22-23: please include that the bacteria isolates were also isolated form agriculture soil.
- Discussion section does not represent properly the discussion but represent the results of As and Hg tolerance.
- Please include some number in the abstract to give the authors some insights on the ability of bacteria exhibit some PGP traits.
Introduction
- Unfortunately, the introduction does not fell in the scope of the manuscript. The introduction is too long and get the readers bored. Indeed, too many sections are not in the current place and must be moved to the discussion section. The appropriate structure of introduction is: the main problem or the issue you are going to treat, then you present or the current solution, and the solution you propose, and finally you present the objectives of your study.
- L42: A verb is missing, I suggest mining activities cause severe…..
- L46: What are other potentially toxic elements than heavy metals released by mining activities?
- L48-L50: in general, the presence of heavy metals in the soil represents a risk for the environment and the human health, and the transportation of heavy metals by the rainfall represent only a part of the consequence of the presence of heavy metals in the soil.
- L54-L58: I found that these sentences are not relevant to your study, and you must focus on the main objective of your study.
- L59-L63: In my opinion, this section is controversial. In the beginning you have stated that the current strategy which could be physical, chemical, and biological. However, you note that these techniques are costly and not effective. But then you have mentioned that phytoremediation and bioremediation, which are biological approaches, are ecologically sound approaches. In the literature, the biological approaches are not costly, environmentally friendly.
- L64-68: I think that this section is not relevant to your study since you did not investigate the tolerance mechanisms.
- L77-78: does IAA, phosphate solubilization, and siderophores production are among the heavy metal tolerance mechanisms?
- L81-94: this section should be moved to the discussion section.
- L95-L110: the section is not relevant to your study, you should focus on bioremediation using bacteria and how the bio-remediation could be achieved through the application of this bacteria
- L118-121: since the mechanisms were not included in the study, this sentence should be removed.

Validity of the findings

Material and methods
- I suggest that the material and methods section should be divided into sub-section such as “site study and soil sampling”, “isolation of cultured rhizobacteria”, “plant growth promoting traits”, and “As-Hg minimal inhibitory concentration” which will facilitate the reading.
- The physico-chemical characteristics of the three soils are missing. At least, the content of heavy metals of the three soils should be given.
- How the soil was sampled, how many repetition for each site.
- The molecular identification of the isolated strain is missing, and the identification based on fatty acids is not sufficient.
- Does the L-tryptophan was added to the medium in case of IAA production, if yes how much ?
- Pikosvskaya is not appropriate for phosphate solubilization, according to many studies NBRIY is more appropriate.
- Which source of phosphate was used to screen the ability of phosphate solubilization.
- In my opinion, you should start with heavy metals tolerance, then the PGP traits.
- The investigation of the tolerance ability in the agar medium is not accurate, since the medium contain agar which could affect the availability of the heavy metals. Instead, I suggest using the broth medium.
Results
- How many bacteria were isolated. I can’t find the correct answer. In the abstract, I found that 99 strains were isolated, and only 48 were identified, and base on what criteria the bacteria were selected. In the results section, I found that 7 strains were isolated from site 1, 19 from site 2, and 21 form site 3, which make a total of 47. Please rectify.
- L204: “Among 27 strains” does only 27 strains were tested for PGP traits. From 47 or 48 strains only 27 were tested for the ability to exhibit PGP traits and how the 27 strains were selected is missing.
Discussion
- Long and broad discussion, you should focus on you results and compare them with other studies.
Conclusion
- L387-389: results of your studies does not support this sentence, since you have isolated some bacteria and you have not consider the bacterial diversity as whole.
- L396-397: which scenario?
- L397-400: results of your study does not support this statement, since future research should focus on the ability of bacteria to remove heavy metals from media, the tolerance mechanisms, and investigate the plant-rhizobacteria under greenhouse experiment before moving to the field.

Additional comments

- I suggest the the following recent and highly impacted studies, which you can cite them into the introduction and discussion section, which I found them relevant to your study:
- Raklami, A., Oubane, M., Meddich, A., Hafidi, M., Marschner, B., Heinze, S., and Oufdou, K. (2021). Phytotoxicity and genotoxicity as a new approach to assess heavy metals effect on Medicago sativa L.: Role of metallo-resistant rhizobacteria. Environmental Technology & Innovation. 24, 101833. doi: 10.1016/j.eti.2021.101833
- El Alaoui, A., Raklami, A., Bechtaoui, N., El Gharmali, A., Ouhammou, A., Imziln, B., Achouak, W., Pajuelo, E., and Oufdou, K. (2021). Use of native plants and their associated bacteria rhizobiomes to remediate-restore Draa Sfar and Kettara mining sites, Morocco. Environmental Monitoring and Assessment. 193, 232. doi: 10.1007/s10661-021-08977-4
- Raklami, A., Tahiri, A. I., Bechtaoui, N., El Gharmali, A., Pajuelo, E., Baslam, M., Meddich, A., and Oufdou, K. (2020). Restoring the plant productivity of heavy metal-contaminated soil using phosphate sludge, marble waste, and beneficial microorganisms. Journal of Environmental Sciences. 99, 210-221. doi: 10.1016/j.jes.2020.06.032
- Raklami, A., Oufdou, K., Tahiri, A., Mateos-Naranjo, M., Navarro-Torre, S., Rodríguez-Liorente, I.D., Meddich, A., Redondo-Gómez, S. and Pajuelo, E. (2019). Safe cultivation of Medicago sativa on metal polluted soils of semi-arid regions assisted by heat- and metallo-resistant PGPR. Microorganisms. 7 (7), 212. doi: 10.3390/microorganisms7070212
- Raklami, A., Meddich, A., Oufdou, K. and Baslam, M., 2022. Plants—Microorganisms-Based Bioremediation for Heavy Metal Cleanup: Recent Developments, Phytoremediation Techniques, Regulation Mechanisms, and Molecular Responses. International Journal of Molecular Sciences, 23(9), p.5031. doi: 10.3390/ijms23095031
- Raklami, A., Meddich, A., Pajuelo, E., Marschner, B., Heinze, S. and Oufdou, K., 2022. Combined application of marble waste and beneficial microorganisms: Toward a cost-effective approach for restoration of heavy metals contaminated sites. Environmental Science and Pollution Research, pp.1-15. Doi: 10.1007/s11356-022-19149-3
- Raklami, A., El Gharmali, A., Ait Rahou, Y., Oufdou, K. and Meddich, A., 2021. Compost and mycorrhizae application as a technique to alleviate Cd and Zn stress in Medicago sativa. International Journal of Phytoremediation, 23(2), pp.190-201. Doi: 10.1080/15226514.2020.1803206

---

## Round 0.2 · accepted · Accept

Dear Dr. Rojas-Solis and colleagues:

Thanks for revising your manuscript based on the concerns raised by the reviewers. I now believe that your manuscript is suitable for publication. Congratulations! I look forward to seeing this work in print, and I anticipate it being an important resource for groups studying plant growth promoting traits and As-Hg tolerance of rhizobacteria from mines. Thanks again for choosing PeerJ to publish such important work.

Best,

-joe

Reviewer 2 ·

Basic reporting

The manuscript is now improved following perfect structure of the article, tables and figures.

Experimental design

The methods are described with sufficient details.

Validity of the findings

The findings comply to the rationale of the study.

Additional comments

The manuscript can be accepted.